# Predicting Silent Atrial Fibrillation in the Elderly: A Report from the NOMED-AF Cross-Sectional Study

**DOI:** 10.3390/jcm10112321

**Published:** 2021-05-26

**Authors:** Katarzyna Mitrega, Gregory Y. H. Lip, Beata Sredniawa, Adam Sokal, Witold Streb, Karol Przyludzki, Tomasz Zdrojewski, Lukasz Wierucki, Marcin Rutkowski, Piotr Bandosz, Jaroslaw Kazmierczak, Tomasz Grodzicki, Grzegorz Opolski, Zbigniew Kalarus

**Affiliations:** 1Department of Cardiology, Silesian Centre of Heart Diseases, 41-800 Zabrze, Poland; bms@pro.onet.pl (B.S.); a.sokal@scs.pl (A.S.); w.streb@sccs.pl (W.S.); k.przyludzki@sccs.pl (K.P.); zbigniewkalarus@kalmet.com.pl (Z.K.); 2Liverpool Centre for Cardiovascular Science, University of Liverpool and Liverpool Heart & Chest Hospital, Liverpool 14 3PE, UK; Gregory.Lip@liverpool.ac.uk; 3Aalborg Thrombosis Research Unit, Department of Clinical Medicine, Aalborg University, DK-9100 Aalborg, Denmark; 4Department of Cardiology, Medical University of Silesia, DMS in Zabrze, 40-055 Katowice, Poland; 5Silesian Park of Medical Technology Kardio-Med Silesia in Zabrze, 41-800 Zabrze, Poland; 6Department of Preventive Medicine and Education, Medical University of Gdansk, 80-210 Gdansk, Poland; trzdroj@gmail.com (T.Z.); lukasz.wierucki@gumed.edu.pl (L.W.); marcin.rutkowski@gumed.edu.pl (M.R.); piotr.bandosz@gumed.edu.pl (P.B.); 7Department of Cardiology, Pomeranian Medical University, 70-204 Szczecin, Poland; jar.kazmierczak@o2.pl; 8Department of Internal Medicine and Gerontology, Jagiellonian University Medical College, 31-007 Krakow, Poland; tomekg@su.krakow.pl; 9First Chair and Department of Cardiology, Medical University of Warsaw, 02-091 Warsaw, Poland; grzegorz.opolski@gmail.com

**Keywords:** silent atrial fibrillation, risk factors, risk assessment

## Abstract

Background: Silent atrial fibrillation (SAF) is common and is associated with poor outcomes. Aims: to study the risk factors for AF and SAF in the elderly (≥65 years) general population and to develop a risk stratification model for predicting SAF. Methods: Continuous ECG monitoring was performed for up to 30 days using a vest-based system in a cohort from NOMED-AF, a cross-sectional study based on a nationwide population sample. The independent risk factors for AF and SAF were determined using multiple logistic regression. ROC analysis was applied to validate the developed risk stratification score. Results: From the total cohort of 3014 subjects, AF was diagnosed in 680 individuals (mean age, 77.5 ± 7.9; 50.1% men) with AF, and, of these, 41% had SAF. Independent associations with an increased risk of AF were age, male gender, coronary heart disease, thyroid diseases, prior ischemic stroke or transient ischemic attack (ICS/TIA), diabetes, heart failure, chronic kidney disease (CKD), obesity, and NT-proBNP >125 ng/mL. The risk factors for SAF were age, male gender, ICS/TIA, diabetes, heart failure, CKD, and NT-proBNP >125 ng/mL. We developed a clinical risk scale (MR-DASH score) that achieved a good level of prediction in the derivation cohort (AUC 0.726) and the validation cohort (AUC 0.730). Conclusions: SAF is associated with various clinical risk factors in a population sample of individuals ≥65 years. Stratifying individuals from the general population according to their risk for SAF may be possible using the MR-DASH score, facilitating targeted screening programs of individuals with a high risk of SAF.

## 1. Introduction

Asymptomatic (‘silent’) atrial fibrillation is common and is associated with poor outcomes. Given the increasing incidence of atrial fibrillation (AF) worldwide, it is important to determine the risk factors for AF occurrence, which will allow for rapid diagnosis among patients who are most predisposed to their development.

The risk factors for AF are well established and include hypertension, heart failure (HF), physical activity, obesity, chronic coronary syndromes, chronic kidney disease (CKD), and hyperthyroidism [1,2,3,4]. The determination of risk factors is particularly important in patients with silent atrial fibrillation (SAF), but their identification is a challenge in everyday clinical practice and, often, such patients are diagnosed for the first time when they present with an AF-related complication, such as stroke or heart failure; however, general population-based data for silent AF (SAF) are limited.

In the EORP-AF registry, SAF occurred more frequently in elderly patients with more comorbidities (e.g., previous myocardial infarction, previous coronary artery bypass graft, chronic kidney disease, or peripheral vascular disease) and conferred a higher thromboembolic risk [5]. Indeed, elderly patients with SAF, and those with SAF associated with CKD or HF, were at a higher risk of death at 1 year compared to symptomatic patients. The aim of the present ancillary analysis from the NOMED-AF study is to report the risk factors for symptomatic AF and SAF in the elderly (≥65 years) general population. Secondly, we aim to develop a risk stratification model for predicting SAF.

## 2. Materials and Methods

The NOMED-AF study was a cross-sectional study based on a nationwide sample of adults aged ≥65 years (n = 3014; mean age 77.5 ± 7.9 years; 50.9% male). A complete description of the study design and methodology has been published elsewhere [6]. The aim of NOMED-AF was to assess the frequency of symptomatic and silent AF and identify their risk profiles in a Polish population of adults aged ≥65 years. The target sample size comprised 3000 individuals, and enrollment was performed between 15 March 2017 and 10 March 2018. All participants were equipped with a continuous electrocardiogram (ECG) recording vest for up to 30 days.

Briefly, recruitment to the study was based on a random sampling of individuals aged ≥65 years, regardless of their health status. The sampling was stratified by age, gender, living environment (urban, rural), and regions (16 districts) to reflect a cross-section of the Polish elderly population. Written informed consent was obtained from all participants.

The NOMED-AF study was approved by the Local Bioethical Committee (26/2015) and registered at clinicaltrials.gov (accessed on 15 March 2017) (NCT03243474). The study procedures consisted of: (i) a survey on either social and health status or medication, (ii) anthropometric measurements (body mass, height, etc.), (iii) blood pressure measurements, and (iv) the collection of biological samples (blood and urine) for further investigation and long-term continuous ECG monitoring in all recruited individuals. All of the above procedures were performed on site during two study visits by a professionally trained study nurse.

### 2.1. ECG Monitoring and Clinical Assessments

All participants were equipped with a continuous electrocardiogram (ECG) recording vest for up to 30 days. They were encouraged to wear the vest for the maximum possible time, with short breaks for either hygiene (washing) or other necessary reasons only. The ECG signal was transmitted to the Integrated Medical Algorithms Platform (Comarch Healthcare S.A., Krakow, Poland). All episodes automatically classified as AF by the platform’s software were verified and finally confirmed as AF by cardiologists. According to the current ESC guidelines [1], only participants with AF episodes lasting longer than 30 s were included in the analysis as AF-positive individuals. Silent AF was defined as AF that was detected and confirmed by cardiologists in asymptomatic individuals. The scheme of the long-term ECG monitoring system is shown in Figure 1.

To determine the possible risk factors, clinical parameters that could have an impact on the occurrence of AF and/or SAF were retrieved for risk stratification analysis. The recorded data included demography, cardiovascular diseases (including myocardial infarction, HF, previous revascularization, hypertension, previous stroke, and thromboembolic disease), metabolic diseases (i.e., diabetes mellitus or thyroid disease), chronic kidney diseases, chronic obstructive pulmonary disease, and relevant biomarkers.

Hypertension (HA) was diagnosed if the average blood pressure values from two measurements during each visit were equal to or higher than 140 mmHg (systolic) and/or 90 mmHg (diastolic), or if the patient had taken antihypertensive drugs in the previous 2 weeks because of an earlier diagnosis of hypertension, as per the 2018 ESC/European Society of Hypertension guidelines for the management of arterial hypertension [7]. Diabetes mellitus (DM) was diagnosed when hemoglobin A1c was ≥6.5%, or if the patient was aware of their diabetes and was taking glucose-lowering agents, in accordance with the 2019 American Diabetes Association and 2019 ESC/European Association for the Study of Diabetes criteria [8,9]. The hemoglobin A1c test was performed using a method certified by the National Glycohemoglobin Standardization Program, certified and standardized to the Diabetes Control and Complications Trial assay. CKD was defined as an estimated glomerular filtration rate <60 mL/min/1.73 m^2^ or ≥60 mL/min/1.73 m^2^ with coexisting albuminuria (albumin-to-creatinine ratio ≥30 mg/g) using the 2009 Chronic Kidney Disease Epidemiology Collaboration formula [10,11,12]. Definitions of these medical conditions are summarized in Appendix A Appendix A (online appendix).

### 2.2. Statistical Analysis

All statistical analyses were performed using SPSS v19 (IBM Corp., Armonk, NY, USA). The risk factor analysis was conducted on a sample weighted to the Polish population, taking into account the complex sampling scheme. The distributions and the 95% confidence intervals of the risk factors were obtained for the AF and SAF groups. The significance was tested by chi^2^ and t-tests for age. Risk factors associated with AF or SAF were first identified using univariate, then multivariate logistic regression. Odds ratios (OR), their significance, and their 95% confidence intervals were obtained.

Based on the recognized risk factors, we constructed a risk prediction scheme for silent atrial fibrillation (MR-DASH score), focused on including only components that are easily accessible as part of patients’ daily routines and which do not require additional testing. Hence, we included all identified factors in our risk scale, omitting high NT-proBNP levels. To create this risk score, the study population without overt AF was randomly divided in proportions of 2/3 to 1/3, as derivation and validation cohorts, respectively. The relative contribution of particular risk factors to total risk and their significance were evaluated on the basis of 2/3 of the randomly divided population using multivariate logistic regression. This procedure was repeated five times to confirm the consistency of the model.

The final score was defined as the sum of the rounded ORs associated with risk factors, which were significant in 4 or 5 draws. For the risk factors that were significant in 3 or less draws, the score was set to one and added to the final score. Subsequently, the clinical risk score was validated on the remaining 1/3 of the divided population. The c-statistics (AUC) and cutoff value for the proposed scale were calculated with the use of receiver operating characteristic (ROC) analysis to obtain the best balance between sensitivity and specificity. This procedure was also repeated five times to confirm the consistency of the model. The cutoff point was established based on the method of minimal distance to the point (0;1). Finally, the positive (PPV) and negative (NPV) predictive values were calculated separately for each of the 5 draws. *p* values < 0.05 were considered statistically significant.

## 3. Results

In this nationwide population sample, we enrolled 3014 subjects (mean age, 77.5 ± 7.9 years; women, 1479), and long-term ECG monitoring was performed in 2974 (98.7%). The median effective time of monitoring (the time of acquisition of an ECG signal with a quality sufficient for analysis) was 23 days, 10 h, and 26 min (range: 12 min–37 days, 19 h 4 min).

The overall number of individuals that appeared to have AF (symptomatic or silent) was 680 (22.6%). Of these, symptomatic atrial fibrillation was present in 401 (13.3%) and 279 (9.3%) had SAF. The detailed demographic and clinical characteristics of the study population are summarized in Table 1.

### 3.1. Independent Risk Factors for Atrial Fibrillation

In the multivariable analysis, age, male gender, coronary heart disease, thyroid diseases, prior ischemic stroke or transient ischemic attack, diabetes, heart failure chronic kidney disease, obesity (BMI > 30), and an NT-proBNP level above the reference range (>125 ng/mL) were independent risk factors for AF (Table 2). Prior revascularization, either percutaneous or surgical, was associated with a decreased risk of AF but not of SAF. Age, male gender, prior ICS/TIA, diabetes, heart failure, chronic kidney diseases, and an NT-proBNP level above the reference range (>125 ng/mL) were risk factors for SAF (Table 2).

### 3.2. Predicting Silent Atrial Fibrillation in Subjects Aged over 65 Years

The results of the risk factor analysis for five subsequent draws and the assigned scores are presented in Table 3. All of the included risk factors were significant in at least one draw. The allocation of points on the scale was based on the consistency and height of the statistical significance and the relative risk contribution of the specified factors. We assigned three points to “age over 75 years” due to its consistently high odds ratios and statistical significance. Two points were assigned to the following: male sex, heart failure, and a history of ischemic stroke/transient ischemic attack (ICS/TIA). One point was assigned to diabetes and chronic kidney disease due to the fact that they had the lowest odds ratios and borderline significance (Table 3). The randomly allocated 1/3 of the study population was used for the validation of the proposed scale (MR-DASH; M—male, R—renal failure, D—diabetes, A—age, S—stroke/TIA, H—heart failure). Regardless of the sample drawn, the calculated C-statistics based on the receiver operating characteristic (ROC) analysis ranged between 0.726 and 0.709 or between 0.730 and 0.678 for the derivation and validation cohorts, respectively. The optimal cutoff point established based on the method of the minimal distance to the point (0.1) was 4.5. the ROC curves obtained for all draws and calculated c-statistics are shown in Figure 2.

The sensitivities, specificities, and positive and negative predictive values of the MR-DASH scale are presented in Table 4.

## 4. Discussion

To the best of our knowledge, the NOMED-AF study is the only large-volume investigation that utilizes continuous, long-term, noninvasive ECG monitoring for the detection of AF. In this analysis from NOMED-AF, our principal findings are as follows: (i) The main risk factors for SAF were age, male gender, prior ICS/TIA, diabetes, heart failure, CKD, and NT-proBNP > 125 ng/mL; and (ii) a simple clinical risk scale (MR-DASH score) was developed, which had a good level of prediction in the derivation cohort (AUC 0.726) and the validation cohort (AUC 0.730).

Due to the applied methodology of NOMED-AF, the obtained data are representative of the entire Polish population over 65 years old. In the present work, we sought risk factors for either overall or silent atrial fibrillation. As mentioned earlier, the method of atrial fibrillation search utilized in this study was based on long-term ECG monitoring. Thus, the AF diagnoses in the subjects studied in our analysis followed the diagnostic criteria of the recently published ESC guidelines [1]. A European Heart Rhythm Association (EHRA) survey showed that the most common method of SAF identification is 24-h Holter ECG records, especially in patients who have experienced a cryptogenic stroke [7]. Although new technologies are emerging to detect arrhythmias, they are not sensitive enough to confirm the diagnosis of AF and if AF is suspected, verification by performing an ECG is necessary [13,14]. When comparing 24-h Holter ECG records with 30-day event-triggered recorders in patients with cryptogenic stroke, the EMBRACE study showed that patients undergoing 30-day triggered monitoring showed a 5-fold higher rate of SAF [8]. A higher prevalence of SAF, between 12.4% and 46%, was even detected using implantable loop recorders (ILRs) in patients after cryptogenic stroke in observations over 1 year or longer. These findings emphasize the need for long-term continuous monitoring in the evaluation of the true prevalence of SAF. In both studies, with the use of ILR, the highest incidence of newly appeared AF episodes were detected in the first 30 days of monitoring. Hence, our continuous ECG monitoring method, lasting up to 30 days, seems to provide reliable data on the overall and silent AF prevalence in the study population. Additionally, the definitions of some medical conditions (diabetes, hypertension, chronic kidney diseases, thyroid diseases) introduced in our analysis were based not only on medical history and prescribed medication, but also on the results of direct diagnostic tests applied to the whole study population and interpreted according to the current guidelines [9,10,11,12,15,16]. Such an approach significantly strengthens the reliability of the analysis performed.

The risk factor profile of overall (overt and silent) AF was consistent with previous data [1,3,17,18,19]. Only hypertension, which is a generally accepted risk factor of atrial fibrillation, was on the borderline of statistical significance, perhaps reflecting the relatively high prevalence of hypertension (79.5%) in subjects without AF (Table 1). Contrary to the overall AF cohort, some factors, such as coronary heart disease (CHD), obesity, and thyroid diseases, were not significantly predictive of SAF risk; however, thyroid diseases displayed marginal *p*-values, which might be due to their reduced power in the SAF group. Until now, it was believed that ‘general’ (i.e., overall) and silent AF share the same risk factors [5,20,21]. However, our observations suggest that the risk profiles of overt and silent AF overlap but are not identical. The differences in the risk factors between overt AF and SAF relate to CHD, obesity, and thyroid diseases. To classify AF as SAF, two conditions must be granted: AF must occur, and it must be asymptomatic. It is possible that, in individuals with obesity, coronary heart diseases, or thyroid diseases, AF symptoms are more frequent. This can be partially explained by the younger age of individuals with the aforementioned conditions. Indeed, younger subjects are more symptomatic than older subjects due to their more efficient atrio-ventricular conduction, leading to a faster heart rhythm and the feeling of palpitations.

Although some risk scores, such as CHARGE-AF, Framingham, ARIC, and CHA_2_DS_2_VASc, may be useful for risk stratification in patients suspected of having AF [5,22], no scale for the assessment of SAF risk has yet been developed. However, our newly developed SAF score is able to select low-risk subjects. Additionally, the results of our ROC analysis suggest the consistently good predictive value of this schema, with c-statistics comparable to other published AF prediction schema [15,23,24,25,26,27,28,29]. Our proposed SAF scores allow for the identification of subjects who do not require active or intense searching for SAF. Thus far, there are no RCT data that might support the hypothesis that screening for SAF might have potential implications on reducing MACE rates in the elderly; however, based on the MR-DASH scale, such a study could be performed. Further validation of this approach to stratification should be performed in everyday clinical practice.

### Strengths and Limitations

This study provides several novel contributions, including the determination of independent risk factors for SAF and the proposal of a simple score (SAF score) to identify asymptomatic individuals with a low risk of SAF. Our scale is not intended for application in high-risk patients (e.g., patients after cryptogenic stroke), which a priori require an active AF search, but rather to identify asymptomatic patients of low risk who do not require active and systematic SAF searching. As shown in this study, our scale’s consistently high negative predictive value makes it a useful tool in such applications.

## 5. Conclusions

SAF is associated with various clinical risk factors in a population sample of individuals ≥ 65 years. Stratifying individuals from the general population according to their risk of SAF may be possible using the MR-DASH score, facilitating targeted screening programs in individuals with a high risk of SAF.

## 6. Clinical Perspectives

### 6.1. Competency in Medical Knowledge

The risk factors of AF were identified. However, the specific risks associated with silent atrial fibrillation in the general population have not been well characterized. The identification of those risk factors would allow us to better delineate the population at risk of SAF.

### 6.2. Competency in Patient Care

The identification of patients with a low risk of silent atrial fibrillation allows us to concentrate population screening in the moderate- and high-risk population. This would facilitate the identification of subjects with SAF and accelerate the introduction of required treatments.

### 6.3. Translational Outlook

The results of our study should provide the rationale for the organization of systemic populational screening for SAF.

## Figures and Tables

**Figure 1 jcm-10-02321-f001:**
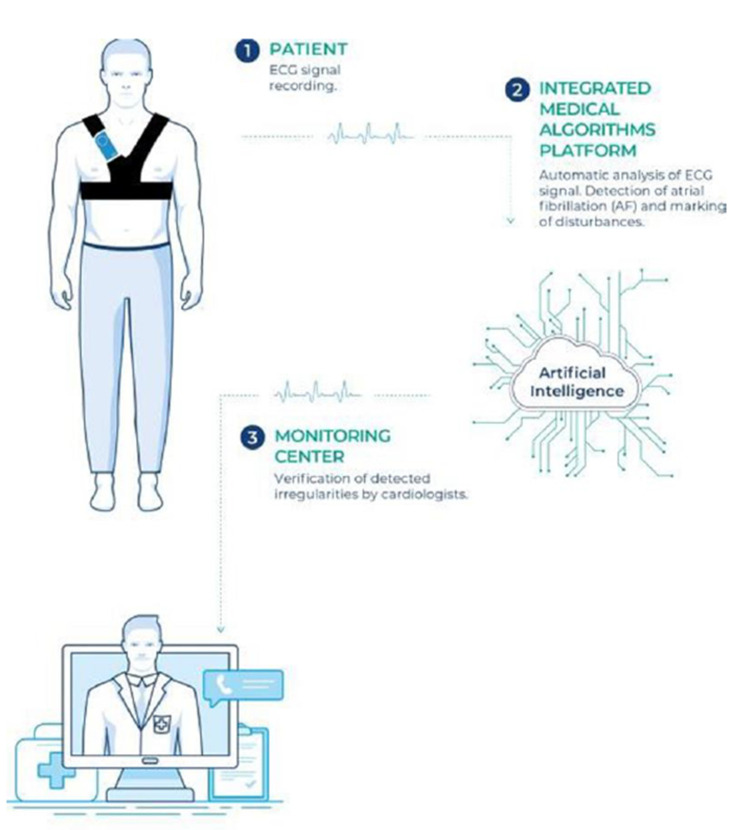
Central illustration: The scheme of the long-term ECG monitoring system used in the NOMED-AF study.

**Figure 2 jcm-10-02321-f002:**
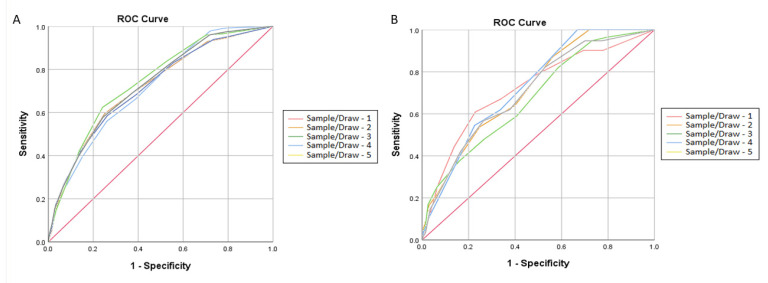
Receiver operating characteristic curves (ROCs) of the proposed risk scale for five random samples obtained in five subsequent draws. C-index (AUC): 95% CI, cutoff point. The cutoff point was determined based on method of minimum distance to the point (0.). **Derivation cohort** (**A**) Sample 1—0.709: 0.672–0.746, 4.5; Sample 2—0.726: 0.687–0.764, 4.5; Sample 3—0.725: 0.689–0.761, 4.5. Sample 4—0.712: 0.675–0.749, 4.5. Sample 5—0.720: 0.683–0.757, 4.5. **Validation cohort** (**B**) Sample 1—0.721: 0.667–0.775, 4.5; Sample 2—0.720: 0.669–0.770, 4.5; Sample 3—0.678: 0.622–0.734, 3.5. Sample 4—0.730: 0.676–0.783, 4.5. Sample 5—0.706: 0.52–0.760, 4.5.

**Table 1 jcm-10-02321-t001:** Demographic and clinical characteristics of the NOMED-AF population.

	Whole Population		No AF (AF−)		AF+			SAF		
	N	%	N	%	N	%	*p* (AF− vs. AF)	N	%	*p* (AF− vs. SAF)
Age (years, mean ± SD)	77.5	7.9	76.8	7.9	80.0	7.4	<0.001	80.9	7.4	<0.001
Male gender	1535	50.9%	1122	48.1%	413	60.7%	<0.001	191	68.5%	<0.001
MI	446	14.8%	321	13.8%	125	18.4%	0.003	45	16.1%	0.294
CHD	666	22.1%	444	19.0%	222	32.6%	<0.001	84	30.1%	<0.001
Thyroid diseases	418	13.9%	301	12.9%	117	17.2%	0.005	42	15.1%	0.337
Pulmonary diseases	361	12.0%	264	11.3%	97	14.3%	0.040	32	11.5%	0.974
Thromboembolism	241	8.0%	166	7.1%	75	11.0%	0.001	23	8.2%	0.513
LEAD	415	13.8%	286	12.3%	129	19.0%	<0.001	51	18.3%	0.005
ICS/TIA	366	12.1%	246	10.5%	120	17.6%	<0.001	53	19.0%	<0.001
PCI/CABG	368	12.2%	270	11.6%	98	14.4%	0.043	44	15.8%	0.040
DM	881	29.2%	628	26.9%	253	37.2%	<0.001	98	35.1%	0.004
Heart failure	673	22.3%	396	17.0%	277	40.7%	<0.001	96	34.4%	<0.001
HA	2433	80.7%	1856	79.5%	577	84.9%	0.001	223	79.9%	0.821
CKD	1005	33.3%	695	29.8%	310	45.6%	<0.001	144	51.6%	<0.001
Physical activity	1294	42.9%	1039	44.5%	255	37.5%	0.001	103	36.9%	0.017
BMI ≥ 30	923	30.6%	686	29.4%	237	34.9%	0.005	88	31.5%	0.419
hs CRP > 5 mg/L	565	18.7%	434	18.6%	131	19.3%	0.659	56	20.1%	0.603
NT pro-BNP > 125 pg/mL	2288	75.9%	1690	72.4%	598	87.9%	<0.001	247	88.5%	<0.001

AF—atrial fibrillation. SAF—silent atrial fibrillation. MI—myocardial infarction CHD—coronary heart disease; LEAD—lower extremity artery disease. ICS—ischemic cerebral stroke; TIA—transient ischemic attack; DM—diabetes mellitus; HA—arterial hypertension; CKD—chronic kidney disease; BMI—body mass index.

**Table 2 jcm-10-02321-t002:** Multivariate analysis of the main risk factors for the occurrence of atrial fibrillation and silent atrial fibrillation.

	AF Overall			SAF		
Parameter	OR	95% CI	*p*	OR	95% CI	*p*
Age (every 5 years)	**1.26**	**1.17–1.35**	**<0.001**	**1.36**	**1.24–1.49**	**<0.001**
Male gender	**2.05**	**1.67–2.51**	**<0.001**	**2.58**	**1.94–3.44**	**<0.001**
MI	0.96	0.70–1.30	0.776	0.68	0.41–1.12	0.131
CHD	**1.30**	**1.01–1.66**	**0.043**	1.10	0.77–1.57	0.592
Thyroid diseases	**1.44**	**1.09–1.90**	**0.010**	1.41	0.98–2.03	0.066
Pulmonary diseases	0.87	0.65–1.17	0.353	0.67	0.43–1.04	0.073
Thromboembolism	1.28	0.91–1.81	0.157	1.41	0.83–2.41	0.204
LEAD	0.96	0.73–1.26	0.761	1.11	0.76–1.62	0.593
ICS/TIA	**1.28**	**1.00–1.64**	**0.051**	**1.59**	**1.16–2.18**	**0.004**
PCI/CABG	**0.43**	**0.30–0.61**	**<0.001**	0.64	0.39–1.06	0.084
DM	**1.39**	**1.12–1.72**	**0.003**	**1.48**	**1.10–1.98**	**0.009**
Heart failure	**2.98**	**2.33–3.80**	**<0.001**	**2.06**	**1.46–2.90**	**<0.001**
HA	1.29	0.97–1.72	0.077	0.85	0.60–1.19	0.344
CKD	**1.25**	**1.00–1.56**	**0.045**	**1.39**	**1.06–1.84**	**0.019**
Physical activity	1.03	0.82–1.29	0.813	1.02	0.78–1.34	0.860
BMI ≥ 30	**1.43**	**1.14–1.78**	**0.002**	1.21	0.92–1.59	0.175
hs CRP >5 mg/L	0.89	0.67–1.19	0.438	0.83	0.59–1.15	0.260
NT pro-BNP > 125 pg/mL	**1.95**	**1.44–2.64**	**<0.001**	**2.37**	**1.493.76**	**<0.001**

AF—atrial fibrillation. SAF—silent atrial fibrillation. MI—myocardial infarction CHD—coronary heart disease; LEAD—lower extremity artery disease. ICS—ischemic cerebral stroke; TIA—transient ischemic attack; DM—diabetes mellitus; HA—arterial hypertension; CKD—chronic kidney disease; BMI—body mass index.

**Table 3 jcm-10-02321-t003:** Rationale for the risk scale for SAF.

Factor		Male Gender	CKD	DM	Age ≥ 75	ICS/TIA	Heart Failure
**Sample/Draw 1**	**OR**	2.39	1.78	1.34	2.66	1.70	1.63
	**95% CI**	1.68–3.38	1.24–2.55	0.93–1.92	1.83–3.87	1.07–2.71	1.09–2.43
	***P***	0.020	0.002	0.115	<0.001	0.025	0.016
**Sample/Draw 2**	**OR**	1.54	1.39	1.39	3.30	1.57	2.02
	**95% CI**	1.07–2.21	0.95–2.05	0.95–2.04	2.21–4.92	0.96–2.57	1.34–3.05
	***P***	<0.001	0.092	0.093	<0.001	0.073	0.001
**Sample/Draw 3**	**OR**	2.13	1.31	1.29	3.10	1.94	2.10
	**95% CI**	1.48–3.06	0.89–1.93	0.88–1.89	2.10–4.57	1.22–3.09	1.40–3.15
	***P***	<0.001	0.171	0.197	<0.001	0.005	<0.001
**Sample/Draw 4**	**OR**	1.99	1.88	1.37	2.34	1.98	1.85
	**95% CI**	1.41–2.81	1.31–2.69	0.96–1.96	1.62–3.38	1.27–3.10	1.26–2.72
	***P***	<0.001	0.001	0.086	<0.001	0.003	0.002
**Sample/Draw 5**	**OR**	2.50	1.43	1.51	2.83	1.76	1.88
	**95% CI**	1.75–3.59	0.98–2.08	1.04–2.20	1.93–4.15	1.09–2.84	1.25–2.83
	***P***	<0.001	0.066	0.030	<0.001	0.021	0.002
**Abbreviation**		**M** **M**ale gender	**R** **R**enal failure	**D** **D**iabetes	**A** **A**ge	**S** **S**troke	**H** **H**eart failure
**Score**		**2**	**1**	**1**	**3**	**2**	**1**

Weight of selected risk factors on the basis of five subsequent random splits of study population in derivation (2/3 of study population) and validation groups (1/3 of study population). ICS/TIA—ischemic cerebral stroke/transient ischemic attack; DM—diabetes mellitus; CKD—chronic kidney disease.

**Table 4 jcm-10-02321-t004:** Sensitivity, specificity, and positive and negative predictive values of designed risk scale, calculated based on five subsequent random splits of the study population in the derivation (2/3 of the study population) and validation cohorts (1/3 of the study population).

	Sample/Draw 1		Sample/Draw 2		Sample/Draw 3		Sample/Draw 4		Sample/Draw 5	
	Derivation Cohort	Validation Cohort	Derivation Cohort	Validation Cohort	Derivation Cohort	Validation Cohort	Derivation Cohort	Validation Cohort	Derivation Cohort	Validation Cohort
Sensitivity (%)	52.2	60.9	59.9	53.8	62.7	48.0	58.3	54.1	57.9	56.9
Specificity (%)	74.0	77.2	74.9	75.3	75.9	73.2	735.7	77.4	75.7	73.7
Positive predictive value (%)	17.2	18.1	16.9	18.9	19.0	143.7	18.2	15.4	18.3	16.1
Negative predictive value (%)	94.6	96.0	95.6	93.8	95.8	93.6	94.6	95.7	95.0	95.1

## Data Availability

Data supported the reported results will be available at jakub@pzh.gov.pl.

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
