# Peer review of "Predicting Silent Atrial Fibrillation in the Elderly: A Report from the NOMED-AF Cross-Sectional Study"

_jcm, 2021, doi:10.3390/jcm10112321_

Round 1
Reviewer 1 Report
Excellent, well-designed study. Novel results including determining of the independent risk factors for silent AF and proposing a risk score to identify asymptomatic individuals with low risk of silent AF. The manuscript is well-written, with strong references.
The topic is of great interest to the readers. The results are of relevance and consistent. The conclusions are practical and important.
Minor language polishment required. There are some typographic errors. There are very long sentences. To improve readability, consider breaking them into multiple sentences. Figure 1 needs to be resubmitted with better quality.
Author Response
Thank you for your revision. The language polishment and readability improvement has been performed. We also resubmitted Figure 1 with better quality.
Reviewer 2 Report
Dear authors,
it was a pleasure to receive your paper entitled “Predicting silent atrial fibrillation in the elderly. A report from the NOMED-AF cross-sectional study „.
Indeed, the general design of the study is good, the quality of the data analyzed seems high and, what is more, the statistical analysis is comprehensive. However, there are some points that need to be addressed:
- The authors state in their discussion that “Contrary to the overall AF cohort, some factors as coronary heart disease (CHD), obesity and thyroid diseases were not significantly predictive of SAF risk“
This is an interesting finding. However, as shown in Table 2: the p-values for PCI/CABG in SAF was 0.084, and for thyroid disease it was 0.066. Both are marginal p-values that could be simply due to reduced power in the SAF group. This needs at least to be discussed.
- The authors suggest risk profiles of overt and silent AF are overlapping but not identical, which is interesting. However, they do not provide any explanation of the potential pathomechanism behind this finding.
- Despite the fact that screening of SAF might have potential implications on reducing MACE rates in elderly. There is a lack of RCT evidence to support this hypothesis thus far. This needs at least to be acknowledged.
Kind regards
Author Response
Thank you for your valuable review and accurate comments.
Referring to:
1) we added in the discussion the sentence cit.:
"Contrary to the overall AF cohort, some factors as coronary heart disease
(CHD), obesity and thyroid diseases were not significantly predictive of SAF
risk, however thyroid diseases were marginal p-values ​​and it might be due to
reduced power in SAF group."
Although the p-value for PCI/CABG was 0.084, we did not take PCI/CABG as
a risk factor for SAF, only as a potential protective factor.
2) We provided an explanation of the potential pathomechanism behind the finding that overt and silent AF are overlapping but not identical: "The differences in risk factors between overt AF and SAF relate to CHD, obesity and thyroid diseases. To classify AF as a SAF, two conditions must be granted: AF must occur and it must be asymptomatic. This is possible that in individuals with obesity, coronary heart diseases or thyroid diseases AF symptoms are more frequent. This can be partially explained by the younger age of individuals with mentioned conditions. Indeed, younger subjects are more symptomatic than older due to more efficient atrio-ventricular conduction leading to faster heart rhythm and feel of palpitation."
3)We added in the discussion the sentence cit.:"So far, there are no RCT data
which might support the hypothesis that screening of SAF might have potential implications on reducing MACE rates in elderly, however, based on the MR-DASH scale, such a study could be performed."
Reviewer 3 Report
In this study, Mitrega et al present an ancillary analysis from the NOMED-AF study aimed at reporting the risk factors for symptomatic AF and SAF in an elderly (≥65 years) general population. Moreover, the Authors developed a risk stratification model for predicting SAF. Overall, the idea behind this study is good in terms of novelty and clinical relevance. The identification of prediction models may represent a step forward in understanding and treating this condition.
The manuscript is well written, with only minor English changes being required.
However, one issue should be better addressed. In particular, the Authors should include information on pharmacological therapies in Table 1. AF has been associated with cardiovascular drugs such as adenosine, dobutamine, and milrinone. In addition, medications such as corticosteroids, ondansetron, and antineoplastic agents as well as a history of alcohol abuse have been reported to induce AF. Even bisphosphonate drugs have been somehow associated with new onset AF. Overall, this information is missing both from the submitted manuscript and from the previously published NOMED-AF study (10.5603/KP.a2018.0193). This information should eventually be taken into account in multivariate analyses.
Author Response
Thank you for your review and valuable comments. Pharmacotherapy
is a subject of separate analyzes. Drugs such as dobutamine, milrinone,
adenosine were not used at all among the patients who participated in the
study. In turn, we do not have data on corticosteroids, antineoplastic
agents and bisphosphonates. Regarding alcohol, we also have no objective
data on alcohol abuse.
English language has been improved by native speaker.